# Programmed Cell Death in the Left and Right Ventricle of the Late Phase of Post-Infarction Heart Failure

**DOI:** 10.3390/ijms21207782

**Published:** 2020-10-21

**Authors:** Martin Lichý, Adrián Szobi, Jaroslav Hrdlička, Jan Neckář, František Kolář, Adriana Adameová

**Affiliations:** 1Department of Pharmacology and Toxicology, Faculty of Pharmacy, Comenius University in Bratislava, Kalinčiakova 8, 83232 Bratislava, Slovakia; martin.lichy9@gmail.com (M.L.); adrian.szobi@gmail.com (A.S.); 2Institute of Physiology of the Czech Academy of Sciences, Vídeňská 1083, 142 20 Prague, Czech Republic; jaroslav.hrdlicka@fgu.cas.cz (J.H.); Jan.Neckar@fgu.cas.cz (J.N.); kolar@biomed.cas.cz (F.K.)

**Keywords:** heart failure, cell death, autophagy, necroptosis

## Abstract

While necroptosis has been shown to contribute to the pathogenesis of post-infarction heart failure (HF), the role of autophagy remains unclear. Likewise, linkage between these two cell death modalities has not been sufficiently investigated. HF was induced by 60-min left coronary occlusion in adult Wistar rats and heart function was assessed 6 weeks later followed by immunoblotting analysis of necroptotic and autophagic proteins in both the left (LV) and right ventricle (RV). HF had no effect on RIP1 and RIP3 expression. PhosphoSer229-RIP3, acting as a pro-necroptotic signal, was increased in LV while deceased in RV of failing hearts. Total MLKL was elevated in RV only. Decrease in pSer555-ULK1, increase in pSer473-Akt and no significant elevation in beclin-1 and LC3-II/I ratio indicated rather a lowered rate of autophagy in LV. No beclin-1 upregulation and decreased LC3 processing also suggested the inhibition of both autophagosome formation and maturation in RV of failing hearts. In contrast, p89 PARP1 fragment, a marker of executed apoptosis, was increased in RV only. This is the first study showing a different signaling in ventricles of the late phase of post-infarction HF, highlighting necroptosis itself rather than its linkage with autophagy in LV, and apoptosis in RV.

## 1. Introduction

Understanding the mechanisms of cell death in heart failure (HF) after myocardial infarction (MI) is of great clinical importance, since the extent of cellular loss significantly determines the degree of cardiac injury, contractile dysfunction, adverse remodeling and finally patient prognosis [1,2,3]. Therefore, an enormous effort has been undertaken to elucidate the role of certain cell death modalities in HF, and current knowledge on autophagy [4,5,6] and necroptosis [4,7,8,9,10], commonly occurring in post-infarction HF [4,5,7,8,9,10], has been advanced.

Necroptosis has been recognized as a necrosis-like programmed cell death mode [11]. Its molecular canonical pathway involves the activation of RIP1-RIP3 complex with the resultant phosphorylation of MLKL. As a result, MLKL aggregates into a cytotoxic homo/hetero-amyloid structure [12,13,14,15] and translocates into the plasma membrane causing the alterations in ion homeostasis, membrane disruption, and finally cell lysis [13,16,17,18]. In addition, MLKL has also been reported to facilitate the proteases-mediated degradation of plasma membrane structures, and promote membrane oxidative stress via NOX1 activation and NLRP3-mediated inflammation [19]. These features have highlighted the role of MLKL in the execution of necroptosis and thus it is considered to be a terminal necroptotic protein. On the other hand, a role of RIP1 in the activation of necroptosis has been challenged and RIP1-independent mechanisms of the RIP3-MLKL interplay in necroptosis execution have also been indicated [19,20]. 

Autophagy has been known as an adaptive catabolic process activated under stressful conditions [21]. In diseased heart, however, it can also be a reason of cellular demise [22]. The initiation of autophagy mostly relies on ULK1 complex activation which can be promoted by AMPK and repressed by the Akt-mTOR axis [23,24]. In the early phase of autophagy, ULK1 complexes with Beclin-1, thus ensuring proper phagophore formation on the endoplasmic reticulum and its expansion into cytosol [25,26]. Once formed, lipidated LC3 incorporation into the autophagic vesicles drives their maturation into autophagosomes which eventually fuse with lysosomes [21,26,27,28]. 

Both necroptosis and autophagy have been studied in post-ischemic HF as stand-alone cell death modalities [5,7,29,30]. However, their possible interweaving in this cardiac syndrome has not been thoroughly studied so far. From the literature, it is known that necroptosis could be linked with autophagy by several ways including necrosome-autophagosome interaction [31] or necroptosis-mediated inhibition of autophagic flux [11,32]. This could suggest that the interconnection of these two cell death modes is of detrimental nature. On the other hand, it seems to be more complicated because while necroptosis has consistently been suggested to underlie some pathomechanisms of heart failure [30,33], it is still an unresolved question whether the activation of autophagy contributes to HF progression or if it acts rather as a protective mechanism against further myocardial damage of cardiomyocytes [29,34]. In fact, some authors consider necroptosis as an upstream inhibitor of autophagic flux in cardiomyocytes, while others position autophagy upstream of RIP1-mediated necroptosis [32,35]. Recently, it has been reported in human HF that autophagy precedes necroptosis, at least in the terminal stage of this cardiac syndrome [4]. In an effort to expand the knowledge about a role of necroptosis and autophagy in the pathology of a late phase of post-infarction HF and to assess how they might be intertwined, we performed this comprehensive, descriptive study with a particular focus on the protein analysis of their signaling pathways. Besides the investigation of the left ventricle (LV) of failing hearts, which is primarily affected due to MI, we also examined the right ventricle (RV) because its functional impairment is a common clinical outcome of LV dysfunction. In addition, because apoptosis has been indicated as an important reason of cardiomyocyte death in HF [36], although recent data seem to undermine its significance [7,8,30,37], we analyzed some markers of this caspase-dependent cell death pathway as well.

## 2. Results

### 2.1. Weight Parameters and Heart Function 

Data obtained from LV catheterization of post-MI rats showed impaired cardiac systolic and diastolic function indicating the development of HF: developed pressure and rates of pressure development and decline were significantly reduced, and end-diastolic pressure was increased compared to Sham group. Lungs weight and heart weight normalized to body weight were significantly increased in HF group (Table 1).

### 2.2. Necroptosis Signaling

A complex profile of proteins involved in the canonical signaling of necroptosis is depicted in Figure 1. Levels of RIP1, an important necroptotic activator, were more robustly expressed in the RV with no significant change due to HF interference (Figure 1B). The levels of RIP3, acting downstream of RIP1 in the necrosome [38], HF altered its levels neither in the LV nor in the RV tissue (Figure 1C). However, HF significantly affected the level of pSer229-RIP3, recognized as one of the key necroptosis markers which directly activate MLKL [39]. The pSer229-RIP3 level was increased in LV but decreased in RV of failing hearts (Figure 1D). A relative RIP3 phosphorylation showed an increase due to HF in LV myocardium only (Figure 1E). Interestingly, the level of MLKL, as a terminal, cytotoxic pores-forming member of necroptotic cascade, was significantly increased in the RV, but not the LV, of failing hearts (Figure 1F).

### 2.3. Autophagy Signaling

Levels of some important positive (AMPK, Beclin-1, ULK1, LC3) and negative (Akt-mTOR axis) regulators of autophagy (including their activated forms and substrates) are depicted in Figure 2 and Figure 3. Abundance of autophagy activating AMPK and its phosphorylation at Thr172 was higher in the RV than in the LV and not significantly changed in HF (Figure 2B,C). There was no difference among groups in expression of Beclin-1, a substrate of AMPK and a component of a protein complex required for phagosome initiation [40], (Figure 2D). Interestingly, another protein closely related to the first phase of autophagy initiation, ULK1 [40], was significantly decreased only in the LV of failing hearts (Figure 2E), while activating phosphorylation of ULK1 at Ser555 was significantly decreased due to HF in both ventricles of the heart (Figure 2F). However, elevated ratio of pSer555-ULK1 to its total protein level, suggesting increased activity rather upstream of ULK1 than kinase itself, was present solely in the LV of HF group (Figure 2G). Being indispensable in the proper autophagosome formation and cargo selection process [41], LC3-I expression was increased due to HF in ventricular tissue (Figure 2H). However, levels of its active lipidated form LC3-II, which is required for autophagosome sequestration and nucleation [41], were not affected by HF, while the ratio of LC3-II to total LC3-I was significantly decreased in the RV of failing hearts (Figure 2I,J).

Regarding another part of autophagy regulation, expression of Akt, upstream of mTOR [42], was unchanged in HF, as well as its phosphorylation at Thr308 (Figure 3B,C), while phosphorylation at Ser473 was increased by HF only in the LV (Figure 3D). While RV expression of mTOR was significantly increased due to HF, its Ser2448 phosphorylation, which is thought to cause autophagy inhibition by repressing ULK1 activity, was significantly downregulated in both ventricles of failing hearts (Figure 3E,F). These changes led to an increased ratio of pSer2448-mTOR to total mTOR solely in failing LV (Figure 3G). Being downstream of mTOR signaling, pSer757 phosphorylation, which is known to inhibit ULK1 autophagic activity [23], was elevated only in the RV of failing hearts, while the ratio of pSer757-ULK1 to total ULK1 remained unchanged (Figure 3H,I).

### 2.4. Apoptosis Signaling

To assess cell death in the ventricles of failing hearts in a more complex view, we also analyzed some well-established markers of apoptosis. Interestingly, despite being unchanged in its zymogen form, significantly increased levels of cleaved caspase-8 were observed solely in the RV of failing hearts (Figure 4B,C). In spite of this change, the degree of cleavage of its downstream substrate caspase-3 [43] was significantly enhanced in both ventricles of failing hearts (Figure 4D,E). However, the terminal apoptotic marker of PARP1 cleavage, as detected by its p89 fragment, was significantly increased by HF only in the RV (Figure 4F). The Bcl-2 to Bax ratio, another pro-apoptotic indicator, decreased significantly due to HF in both LV and RV (Figure 4G).

## 3. Discussion

In recent years, knowledge on necroptosis has advanced and a practically unknown process has become a well-established pathophysiological factor contributing to myocardial damage in various types of HF [4,7,9,10,30]. In our previous study, we have shown the activation of canonical necroptosis signaling being implicated in both infarcted and non-infarcted LV areas of failing rat hearts, while its execution mechanisms were restricted to the former area only. Such activation of necroptotic signaling has been suggested to be associated with myocardial fibrosis, ventricular remodeling, inflammation and severe contractile dysfunction [8]. Likewise, studies employing explanted human failing hearts of various etiologies have shown the participation of necroptosis in HF-associated myocardial deterioration [4,7]. Some of these studies have also investigated autophagy and reported contradictory data on necroptosis-autophagy link in the heart [4,7,44]. They have assessed necroptosis and autophagy in LV of post-infarction HF only. In this study, we have provided for the first time a complex analysis of signaling molecules of necroptosis and autophagy in both ventricles of rat failing hearts as a dysfunctional RV is a common clinical consequence of left-sided HF. We have assessed a potential interconnection of these cell death modes and their influence on the development of the impaired cardiac function and ventricular remodeling in the late stage of HF following acute MI (42 days after MI). In our previous study employing the same protocol of the coronary artery ligation, it has been reported that the progressive ventricular dilatation develops over the time, but fractional shortening markedly dropped already on day 7 post-infarction with almost no further decline till day 28 [45]. To provide a more comprehensive picture of cell-death in such a model of post-myocardial infarction HF, we have also investigated apoptosis, the relevance of which in this cardiac syndrome remains under scrutiny [7,36,37]. We have found the expression of RIP1 to be significantly increased in the RV, what seems to occur independently of molecular events induced by HF. Although the increased expression of RIP1 has been associated with necroptosis [12], it should also be mentioned that RIP1 has been shown to be dispensable for its activation after certain stimuli [20]. Indeed, RIP3, rather than RIP1 itself, has been suggested to be a crucial regulator processing necroptotic signaling [39]. Although we have found no changes in total RIP3 expression among the groups, its Ser229 phosphorylation, a specific pro-necroptotic phosphorylation, has been increased in the LV, but not in the RV of failing hearts. Because of methodological issues, as addressed in our most recent study [8], as well as discussed elsewhere [19], we assessed only total levels of MLKL instead of p-MLKL, a downstream target of pSer229-RIP3. Expression of MLKL in the LV was not affected by HF, but statistical analysis revealed significantly higher expression of MLKL in the RV of failing hearts. Importantly, as highlighted in the guidelines on investigations of cell death in the myocardium [19], although such an increase in cellular MLKL content can be viewed as an indicator of pro-necroptotic environment, it does not necessarily imply necroptosis execution. To gain more insight into the possible execution mechanisms of necroptosis in the RV of post-ischemic HF, additional experiments evaluating, e.g., MLKL translocation, and/or its post-translational changes such as pSer229-RIP3-dependent phosphorylation, are needed. In spite of this experimental limitation, considering the decreased pSer229-RIP3 levels in the RV, it can be suggested, that despite MLKL upregulation, the post-MI pro-necroptotic environment in the LV spared the neighboring RV, at least in this stage of HF. Such an observation is in accordance with recent knowledge about necroptosis regulation, where particular changes in pSer229-RIP3 corresponded with p-MLKL, thereby indicating the presence or absence of necroptosis [7,8,13,15].

Regarding the relevance of other cell death modalities in late stage of HF, it is still unclear whether autophagy represents the salvage pathway [29] or rather shapes the intracellular environment in favor of cell death execution [4]. Indeed, in post-MI HF, enhanced autophagy has been identified in the end-stage of disease and was linked with an electrical instability of failing myocardium [4,35], but also has been associated with an improvement of cardiac function and remodeling [5]. In our study, autophagy has not been significantly activated, and it seems to be rather inhibited in both ventricles of failing hearts, though its particular mechanisms appear to be tissue dependent. Namely, a decreased LC3-II/I ratio and no beclin-1 upregulation in the RV in HF suggest the inhibition of both autophagosome formation and maturation. Furthermore, the RV in failing hearts has demonstrated alterations in ULK1 signaling consistent with this claim. Specifically, ULK1, a crucial autophagy-initiating regulatory kinase positioned upstream of LC3 and beclin-1 [23], had its inhibitory Ser757 phosphorylation increased while the opposite was true for Ser555, an mTOR-mediated activating phosphorylation [23,24]. In the LV, a depression of pSer555-ULK1 expression comparable to that of the RV together with an increase in pSer473-Akt and no significant elevation in either beclin-1 or LC3 processing have also indicated a lowered rate of autophagy due to HF. These findings are in contrast with some animal and human studies employing a protocol of acute MI [35] or HF [4,5], where LC3 processing was highly elevated.

From aforementioned discussion it can be observed that there have been certain discrepancies in ULK1 regulation by AMPK and mTOR. The first irregularity is the lack of congruence between pSer2448-mTOR, an upstream inhibitor of autophagy through ULK1 Ser757 phosphorylation, and pSer757-ULK1 both in the RV and LV. Possibly, either pSer2448-mTOR does not always reflect mTOR activity [42,46,47], or another kinase could target this ULK1 residue in the failing myocardium. Secondly, the AMPK-ULK1 axis, activation of which is recognized as a pro-autophagic signal by antagonizing the effects of mTOR, seems decoupled [23]. Unlike what could be anticipated [48], in spite of unchanged AMPK activation as measured by Thr172 phosphorylation, its autophagy-stimulating Ser555 phosphorylation of ULK1 was decreased in both ventricles of failing hearts. Therefore, it can be hypothesized that in HF Ser555 of ULK1 could be targeted predominantly by other kinases, such as p38β MAPK as has been reported in mouse skeletal muscle [49].

Both necroptosis and autophagy could regulate each other in an upstream-downstream manner. In fact, in some cells, including H9c2 cardiomyocytes, necroptosis has been shown to act as an inhibitor of autophagic flux by “throwing a wrench” into the interaction of autophagosomes with lysosomes [32] or by interfering with LC3-II signaling [50]. On the contrary, in myocytes from neonatal rat hearts autophagy appears to be positioned upstream of necroptosis [35]. This hypothesis is further supported by several other studies of non-cardiac cell lineages which have described autophagy preceding necroptosis [51,52]. According to our best knowledge, there are only three studies investigating such interactions in MI and post-MI HF [4,5,44], but they have focused only on LV analysis. Furthermore, while it seems that increased autophagy might correlate with necroptosis in post-ischemic HF in rat [5] or in failing human hearts in the terminal stage [4], a very recent study suggests autophagy as a rather cardioprotective mechanism inhibiting necroptosis [44]. The comprehensive profile of autophagic and necroptotic proteins in both LV and RV of post-MI HF reported herein indicates a different story. In contrast with published research on this phenomena [4,5], our results suggest that necroptosis and autophagy are unlikely to cooperate in driving adverse structural changes and dysfunction observed in the failing myocardium, as no positive association between expression of their molecular markers have been detected. However, it should be mentioned that such cooperation cannot be ruled out in other settings. Indeed, it might have taken place earlier into reperfusion, as a gradual decline in autophagic activity following its primary rise as a reaction to acute myocardial damage shortly after MI was reported in a study by Wang et al. [6]. This might explain the lack of autophagy but increased pro-necroptotic environment in the LV of failing hearts 6 weeks after MI. Certainly, these discrepancies may stem from a different approach to evaluate cell death modalities, as we focused on pSer229-RIP3 as a specific marker associated with necroptosis and inflammation [20,53,54,55], while other groups [4,5] used total levels of RIP1 and RIP3 for necroptosis evaluation, which however are not considered to be specific markers of necroptosis [19]. Moreover, other protocol-specific issues, such as the length of survival period or duration of coronary ligation procedure could also underlie such inconsistent data. At this point, it could also be mentioned that a relationship between necroptosis and autophagy in post-infarction HF may be significantly influenced by myocardial inflammatory status. Extensive necrosis (and maybe also necroptosis) in ischemic cardiomyocytes in the infarcted myocardium activates the innate immune response triggering inflammatory reaction. Infiltration of the infarct with neutrophils, monocytes and their descendant macrophages, dendritic cells, and lymphocytes serves to clear the wound from dead cells, stimulate reparative mechanisms and thereby inducing adverse structural and electrical remodeling [56,57]. Thus, on the one hand, it is proposed that the particular infiltrating cells can underlie and/or contribute to the deleterious pro-inflammatory environment to further promote the damage of cardiac cells. On the other hand, such produced inflammation is integral to tissue repair after MI. Therefore, in this regard, the activation of necroptosis during post-infarction HF can be considered to function equally as necrosis and promote the damage while autophagy can take part in heart protection against MI by degradation of matrix debris and allowing to adapt the immune and cellular responses to the inflammation stress.

Recent studies reported a reduced [4,7] or unaltered apoptotic activation in end stages of human HF [7]. In our study, pro-apoptotic proteins, such as csp-8, csp-3 and Bcl2/Bax ratio indicated some apoptotic activity in both RV and LV of failing hearts (with some minor differences). Interestingly, in spite of being increased in both HF ventricles, only in the RV did the activation of executioner csp-3 materialize in PARP1 cleavage, an unambiguous marker of apoptotic execution. Such apoptotic regulation could imply apoptosis rather than necroptosis or autophagy as the underlying mechanism of progressive RV deterioration triggered by LV damage, at least in this stage of HF. However, in line with recent studies, no significant upregulation of pro-apoptotic PARP1 processing was detected in the LV of failing hearts [4,7,37]. Furthermore, the anomaly between cleavage of csp-3 and its downstream PARP1 in the LV may suggest some distinct non-apoptotic role of csp-3 in myocardial signaling due to HF [58]. Indeed, it has been reported that csp-3 activation in cardiomyocytes could favor the cleavage of cytoplasmic proteins (such as α-actin or troponin T) over nuclear fragmentation [59]. As result, myocytes exhibit severe functional abnormalities and lack of contraction, but do not die of apoptosis, as evidenced in the LV of failing hearts.

## 4. Materials and Methods

### 4.1. Experimental Groups

Adult male Hannover Sprague-Dawley rats (250–300 g, IKEM, Prague, Czech Republic) were housed in groups of 3–4 animals under standard conditions in a room with a constant 12 h:12 h light/dark cycle (lights on 6:00) and a temperature of 22 °C. Animals were fed with a standard pelleted diet and tap water ad libitum.

Rats were randomly assigned into two groups: a group subjected to myocardial infarction with subsequent development of heart failure and a group which underwent a sham procedure (Sham).

### 4.2. Animal Model of Post-Myocardial Infarction Heart Failure and Study Design

Protocol of the study (No. 1037/15-221, 12/1/2018) has been approved by the Ethics Committee of the Faculty of Pharmacy, Comenius University (Figure 5). All described procedures were performed in accordance with the Guide for the care and Use of Laboratory Animals, published by the US National Institutes of Health (NIH publication No 85–23, revised in 1996), preceding an authorization by the Animal Care and Use Committee of the Institute of Physiology of the Czech Academy of Sciences (No. 76/2016).

In anesthetized open-chest animals (sodium pentobarbital 60 mg/kg *i.p.*) myocardial infarction was induced by 60-min ligation of the left coronary artery 1–2 mm distal to the left atrial appendage. In our hands, this procedure inducing irreversible ischemia affects around 40% of LV myocardium in Sprague-Dawley rats [60]. Sham-operated rats (*n* = 6) underwent chest surgery without the occlusion. After the release of the suture and chest closure, all spontaneously breathing animals were allowed to recover from anesthesia in separate cages and were given analgesia (ibuprofen, 20 mg/day *p.o.*) for 3 days. Mortality of rats with MI was 40%. At the end of a 6-week period, the animals were subjected to invasive cardiac function assessment and thereafter sacrificed by exsanguination under anesthesia (Figure 5); blood samples were taken from the right ventricular cavity and hearts were rapidly excised and washed in ice-cold phosphate-buffered saline (PBS). In both experimental groups, free walls of the left and right ventricle were harvested and stored at −80 °C till further molecular analysis. In the HF group (*n* = 9), it was first separated the fibrotic infarcted zone from the remote/non-infarcted zone of the LV which was taken for analyses. In our recent studies using similar protocols, the mean infarcted zone occupied between 30% and 40% of failing LV midwall circumference [8,45,60].

### 4.3. Invasive Cardiac Function Assessment

Six weeks after the surgical procedure, anesthetized spontaneously breathing rats (2% isoflurane in room air) placed on a heating pad keeping the animal core temperature at 37 ± 0.5 °C were subjected to the LV catheterization through the right carotid artery using the SPR-407 microtip pressure transducer. Data were acquired using MPVS 300 system (Millar, TX, USA) connected to PowerLab 8/30 (ADInstruments, Oxford, UK). End-diastolic pressure, developed pressure, peak rates of pressure development and decline and heart rate were averaged from 3 measurements each comprising 5 consecutive cardiac cycles.

### 4.4. SDS-PAGE and Immunoblotting

Necroptosis and autophagy were analyzed in both LV and RV separately according to the Guidelines for evaluating myocardial cell death [19] and a qualitative and semiquantitative approach for cell death modalities by SDS-PAGE and Western blotting was applied [7]. After electrophoresis, proteins were transferred onto PVDF membranes (Immobilon-P, Merck Millipore) and incubated with primary antibodies against Bax (ab182734, Abcam), Bcl-2 (SAB4500003, Sigma-Aldrich), Caspase-3 (#9662, Cell Signaling Technology), Caspase-8 (#4790, Cell Signaling Technology), Cleaved PARP1 p25 (ab32064, Abcam), RIP1 (#3493, Cell Signaling Technology), RIP3 (#15828, Cell Signaling Technology), pThr231/Ser232-RIP3 (#57220, Cell Signaling Technology; corresponds with pThr228/Ser229-RIP3 in rat), MLKL (MABC604, Merck), pan-Akt (#4691, Cell Signaling Technology), pSer473-Akt (#4060, Cell Signaling Technology), pThr308-Akt (#4056, Cell Signaling Technology), AMPK (A3730, Sigma-Aldrich), pThr172-AMPK (#2531, Cell Signaling Technology), mTOR (#2983, Cell Signaling Technology), pSer2448-mTOR (ab109268, Abcam), ULK1 (#8054, Cell Signaling Technology), pSer757-ULK1 (#14202, Cell Signaling Technology), pSer555-ULK1 (#5869, Cell Signaling technology), LC3A/B (#12741, Cell Signaling Technology), Beclin-1 (ab32064, Abcam). Subsequently, membranes were incubated with appropriate HRP-conjugated secondary antibodies, specifically: donkey anti-rabbit IgG (711-035-152, Jackson Immunoresearch) or donkey anti-rat IgG (112-035-175, Jackson Immunoresearch). Signals were detected using enhanced chemiluminescence (Crescendo Luminata, Merck Millipore) and captured by a chemiluminescence imaging system (myECL imager, Thermo Fisher Scientific). Total protein staining of membranes with Ponceau S assessed by scanning densitometry was used as the loading control. Relative expression of protein bands was calculated by normalizing the intensity of a protein band with its whole lane protein staining intensity [8].

### 4.5. Statistical Analysis

The data and statistical analysis comply with the recommendations on experimental design and analysis [61]. Data analyses were performed in a blinded fashion. Group sizes were chosen based on our previous experiences with the model of HF after MI [8,60,62]. Data are expressed as mean ± SEM for the number (*n*) of animals in the group. Two-way ANOVA (2WA) coupled with a post-hoc test (Sidak’s multiple comparisons test) were applied for the comparison of differences in variables with normal distribution among 4 groups with the two 2WA factors being “ventricle” (LV-RV) and “heart failure” (Sham-HF). Differences between two groups were compared with an unpaired t-test. All analyses were performed with GraphPad Prism (version 7.00, GraphPad Software, Inc., CA, USA) and differences between groups were considered significant when *p* < 0.05.

## 5. Conclusions

In conclusion, the LV of failing hearts, where myocardium suffered a significant damage by MI, displayed increased pro-necroptotic environment as represented by upregulation of pSer229-RIP3 (Figure 6) and supported by a detailed analysis reported elsewhere [8]. On the other hand, autophagy-associated cell death does not seem to have a substantial effect in promoting the functional deterioration of the LV, as we have found no evidence of its activation under these settings (Figure 6). This seems to rule out the possibility that autophagy is co-stimulated alongside necroptosis in dysfunctional cardiomyocytes, at least in this stage of disease progression. Furthermore, it is questionable whether LV tissue injury secondarily alters the survival of cardiomyocytes in the RV by affecting necroptosis, autophagy or co-stimulation of both, as their key readouts (pSer229-RIP3, LC3 processing) showed HF-associated downregulation. While suppressed apoptosis in the LV of HF is in-line with previously published studies, this cell death mode could be responsible for deteriorating RV functionality following post-infarction LV damage (Figure 6). Taken together, these results highlight a role of necroptosis, but not autophagy, in the pathophysiological mechanisms of HF and underscore the clinical potential of pharmacological interventions aimed at selectively inhibiting the former cell death as a way to improve effectiveness of HF therapy.

## 6. Study limitations

Despite novel information about three major signaling pathways associated with death of cardiac cells in both ventricles of failing hearts, we are aware of some limitations of our study. Firstly, the study is mainly of a descriptive character, lacking some pharmacological interventions and/or genetic modulations specifically targeting key signaling events of the investigated pathways. Secondly, the presented statements rely mostly on the results of extensive protein analysis of signaling molecules. A more detailed insight into key events such as RIP3-MLKL interaction and translocation within cellular membranes and/or autophagosome formation would be supportive for these findings. Likewise, additional analyses—e.g., (immune)histological assessment of the extent of the heart injury as well as a participation of certain infiltrating cells—could provide further insights into the proposed relationship between necroptosis and autophagy and pathomechanisms of a late phase of post-infarction of heart failure. Lastly, our data are attached only to one timepoint of HF, while it develops through various stages over the long time course. Therefore, another set of experiments, with the investigation of hearts from different HF stages (ideally to mimic human HF of NYHA-I to NYHA-IV) would be desirable in order to fully understand relevance of the investigated mechanisms and their interconnection in disease progression.

## Figures and Tables

**Figure 1 ijms-21-07782-f001:**
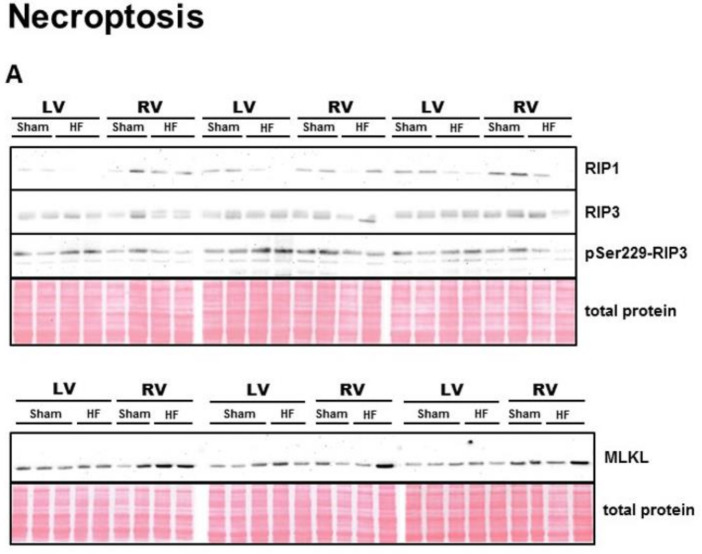
Analysis of necroptotic signaling in the left and right ventricle of rat hearts. (**A**) representative immunoblots and total protein staining; (**B**–**F**) Immunoblot quantification of RIP1, RIP3, pSer229-RIP3, ratio pSer229-RIP3/RIP3 and MLKL. Sham—sham-operated group; HF—group with post-myocardial infarction heart failure; LV—left ventricle; RV—right ventricle. Data are presented as mean ± SEM; *n* = 6–9 per group; 2WA—2-way ANOVA; “HF” factor—presence of heart failure; “V” factor—heart ventricle; “Interaction” of 2 factors. * *p* < 0.05 vs. Sham.

**Figure 2 ijms-21-07782-f002:**
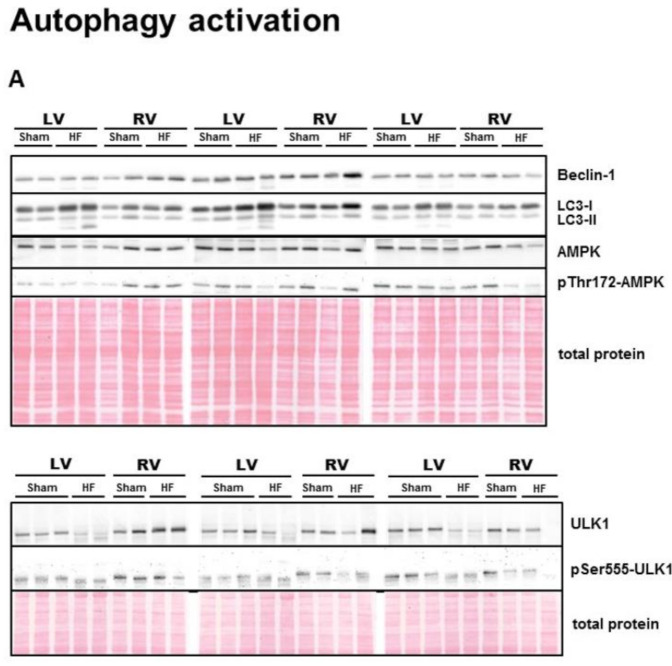
Analysis of activation of autophagic signaling in the left and right ventricle of rat hearts. (**A**) representative immunoblots and total protein staining; (**B**–**J**) Immunoblot quantification of AMPK, pThr172-AMPK, Beclin-1, ULK1, pSer555-ULK1, ratio pSer555-ULK1/ULK1, LC3-I, LC3-II, ratio LC3-II/LC3-I. Sham—sham-operated group; HF—group with post- myocardial infarction heart failure; LV—left ventricle; RV—right ventricle. Data are presented as mean ± SEM; *n* = 6–9 per group; 2WA—2-way ANOVA; “HF” factor—presence of heart failure; “V” factor—heart ventricle; “Interaction” of 2 factors. * *p* < 0.05 vs. Sham.

**Figure 3 ijms-21-07782-f003:**
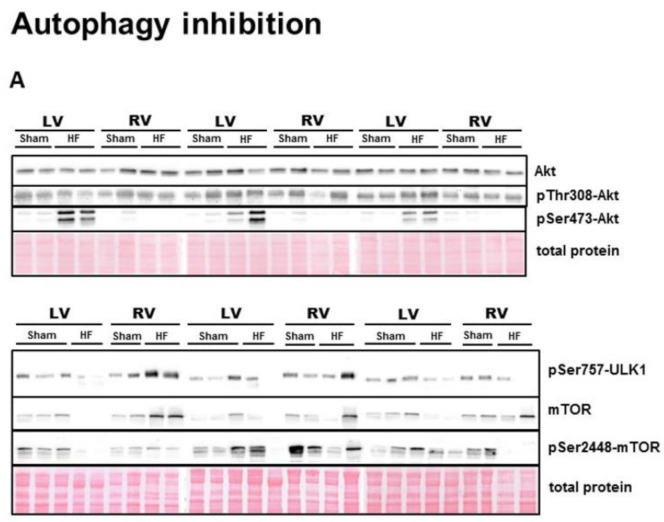
Analysis of inhibition of autophagic signaling in the left and right ventricle of rat hearts. (**A**) representative immunoblots and total protein staining; (**B**–**I**) Immunoblot quantification of Akt, pThr308-Akt, pSer473-Akt, mTOR, pSer2448-mTOR, ratio pSer2448-mTOR/mTOR, pSer757-ULK1, ratio pSer757-ULK1/ULK1. Sham—sham-operated group; HF—group with post-myocardial infarction heart failure; LV—left ventricle; RV—right ventricle. Data are presented as mean ± SEM; *n* = 6–9 per group; 2WA—2-way ANOVA; “HF” factor—presence of heart failure; “V” factor—heart ventricle; “Interaction” of 2 factors. * *p* < 0.05 vs. Sham.

**Figure 4 ijms-21-07782-f004:**
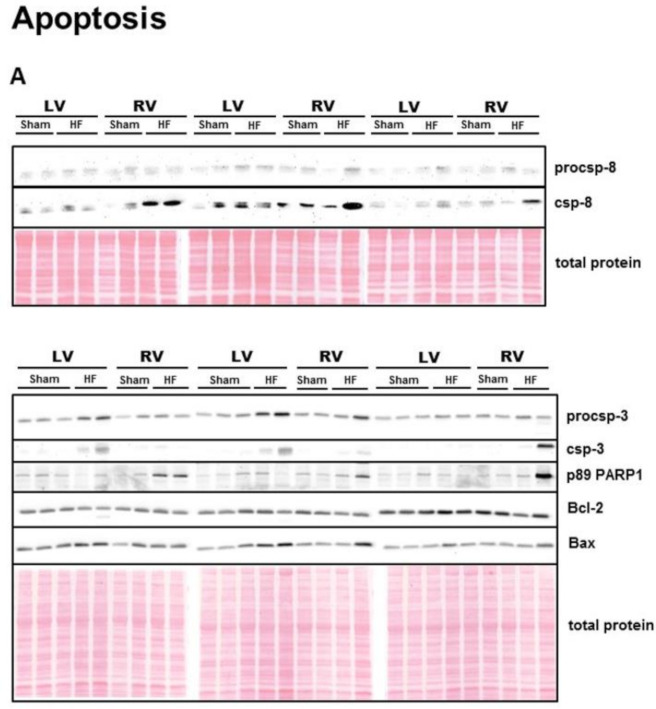
Analysis of apoptotic signaling in the left and right ventricle of rat hearts. (**A**) representative immunoblots and total protein staining; (**B**–**G**) Immunoblot quantification of procsp-8, csp-8, procsp-3, csp-3, p89 PARP1, ratio Bcl-2/Bax in Sham—sham-operated group; HF—group with post-myocardial infarction heart failure; LV—left ventricle; RV—right ventricle. Data are presented as mean ± SEM; *n* = 6–9 per group; 2WA—2-way ANOVA; “HF” factor—presence of heart failure; “V” factor—heart ventricle; “Interaction” of 2 factors. * *p* < 0.05 vs. Sham.

**Figure 5 ijms-21-07782-f005:**
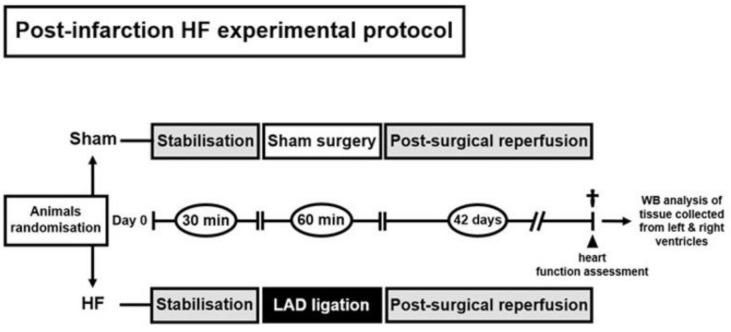
Illustration showing experimental protocol. Sham—group subjected to surgery without coronary occlusion; HF—group subjected to 60 min coronary occlusion with subsequent reperfusion and development of contractile dysfunction and remodeling over the course of 42 days.

**Figure 6 ijms-21-07782-f006:**
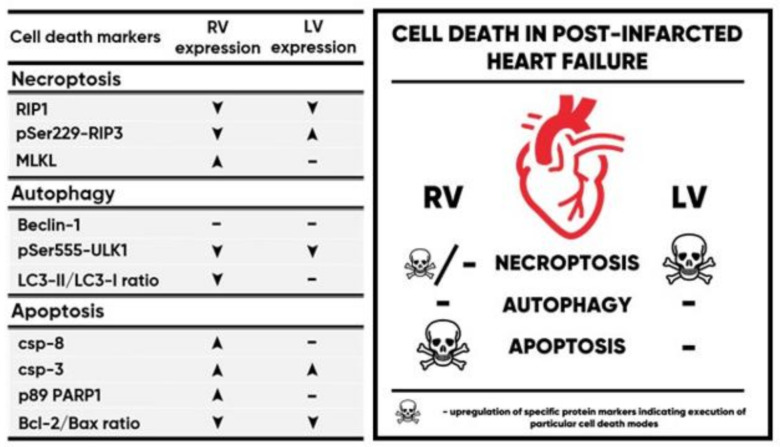
Summary table of selected protein markers of apoptosis, autophagy and necroptosis in the right (RV) and left ventricle (LV) of post-myocardial infarction failing hearts and a schematic diagram showing a relevance of a particular cell death mode in these individual ventricles. ▲/▼—increase/decrease in protein expression or protein ratio.

**Table 1 ijms-21-07782-t001:** Weight parameters and left ventricular function of sham-operated and failing hearts.

	Sham-Operated	Heart Failure
BW (g)	519 ± 22	510 ± 26
HW (mg)	1.40 ± 0.08	1.63 ± 0.09
LW (mg)	1.52 ± 0.05	3.78 ± 0.65 *
HW/BW (mg/g)	2.69 ± 0.08	3.23 ± 0.24 *
LW/BW (mg/g)	2.94 ± 0.08	7.53 ± 1.33 *
LVDevP (mm Hg)	116 ± 5	87 ± 5 *
LVEDP (mm Hg)	9.0 ± 1.5	32.1 ± 4.3 *
+(dP/dt)max (mm Hg/s)	6349 ± 400	3990 ± 306 *
−(dP/dt)max (mm Hg/s)	5613 ± 589	3127 ± 354 *
HR (beats/min)	333 ± 13	318 ± 9

BW, body weight; HW, heart weight; LW, lungs weight; HW/BW, relative heart weight; LW/BW, relative lungs weight; LVDP, left ventricular developed pressure; LVEDP, left ventricular end-diastolic pressure; +(dP/dt)max, maximum rate of pressure development; -(dP/dt)max, maximum rate of pressure decline; HR, heart rate. Data are presented as means SEM, *n* = 6–9 per group. * *p* < 0.05 vs. Sham.

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
