# Peer review of "Programmed Cell Death in the Left and Right Ventricle of the Late Phase of Post-Infarction Heart Failure"

_ijms, 2020, doi:10.3390/ijms21207782_

Round 1
Reviewer 1 Report
Minor comments:
- introdution should be more focused
- "a priori" sample size calculation should be provided
- why lidocaine was not used during surgery to reduce the rate of malignant arrhyrhmias?
Author Response
Dear dr. Bernátová and dr. Barteková,
Thank you for your email dated Sept 21, 2020. We would like also to thank the reviewers for their valuable comments and for giving us an opportunity to revise the manuscript. Please be informed that all comments have been addressed and the manuscript has been revised accordingly. Although it would seem that some points would have benefitted from the supportive data of additional experiments (as might be indicated from the comments of the reviewer No. 2) due to a specific study design, different approach in looking at study endpoints, lack of tissue as well as a short time given us for revision we have addressed some of them by reporting our previous studies and experiences. As can be further seen in the revised text, we have added several new statements and references and the reference section has been altered accordingly. Please be informed that all these changes have been done in a text prepared by the authors, not in a form prepared by the journal office because the additional references added by the EndNote were not able to follow the reference order. The modified parts are highlighted with blue and green color to address the questions of reviewer No.1 and No.2, respectively. In addition, the text editing, for both English language and typographical corrections, has been performed. Please see the revised text.
We believe that the changes made have improved the quality of our article and hope it will meet with your approval.
Reviewer 1
- introdution should be more focused
Please see revised Introduction where some sections have been improved accordingly.
- "a priori" sample size calculation should be provided
The animal study has been conducted in compliance with the recommendations on experimental design and analysis which declares that “the group sizes must be sufficient to permit statistical analyses and 5 is the minimum ‘n’ required for datasets” [1]. In addition, to set an appropriate group size, the estimated mortality of 40-45%, based on our previous studies [2-4], has been taken into consideration while designing the study. Thus, we employed 21 animals totally. Due to experimental losses in the MI group caused by 40% mortality, 15 animals were used for molecular analyses finally (n=6 for Sham and 9 for HF). We have justified the groups size and statistical analysis in subsections of Methods.
- why lidocaine was not used during surgery to reduce the rate of malignant arrhyrhmias?
Lidocaine, as an antiarrhythmic agent, has been shown to exert additional cardioprotective effects under ischemic conditions, e.g. reduce infarct size, cell death, and increase cardiac force [5-7]. Thus, it is likely that this drug can modulate one or more signaling pathways of these outcomes of the heart injury. Because we aimed to investigate molecular signaling of three forms of cell death, and cardiac performance we have designed a study without lidocaine treatment to avoid a possible misinterpretation of findings. This approach is appreciated in both acute as well as chronic ischemia (e.g. post-MI heart failure) which was used in our hands.
Reviewer 2
- The authors examine multiple cell death pathways by blotting for markers of cell death at one time point post cardiac injury. The authors fail to report histological analysis of the tissue examining the amount of cell death that is actually taking place at the time point selected.
As states in the manuscript, the study has been conducted to provide mainly a descriptive protein analysis of molecular signaling of certain cell death forms. Thus, the tissues were handled in a way to follow protocols of biochemical experiments. On the other hand, we are aware that for a detailed picture of certain cell death modalities further analyses by using other methods (including histological analyses) besides western blot would provide supportive data on the amount of cell death. To follow this approach, parallel experimental groups aimed for histological assessment should be employed. Unfortunately, we did not have enough animals for such parallel study and thus not performed histological analyses. On the other hand, similar protocols have been used in our recent studies reporting that the mean infarcted zone occupied between 30% and 40% of failing left ventricle midwall circumference [4, 8]. This information has been added to the methodical section of the revised manuscript and not performing additional, more detailed histological analyses has been indicated as a limitation of the study. In spite of this, as no similar study investigating RV and LV separately and no link between autophagy and necroptosis in this type of HF has been reported so far we believe that the study brings new insights into pathogenesis of heart failure post-myocardial infarction.
- This study would have benefited from the addition of multiple time points to longitudinally assess markers of cell death that change during the progression of heart failure. In addition, cardiac function and dimensions should also be assessed multiple times to correlate with the changes in cell death markers.
As was indicated before in the section of Study limitation “Lastly, our data are attached only to one time point of HF, while it develops through various stages over the long time course. Therefore, another set of experiments, with the investigation of hearts from different HF stages (ideally to mimic human HF of NYHA-I to NYHA-IV) would be desirable in order to fully understand relevance of the investigated mechanisms and their interconnection in disease progression“ we have been aware and fully agree with the reviewer that additional (earlier) time points analyzing cell death markers and heart function would be desirable to reveal longitudinal changes during the development and progression of heart failure. However, in this particular study we have focused on a late phase of post-infarction HF only. Therefore, we believe that it can be considered as a pilot study in this research area and a more comprehensive study employing various time points of disease progression can be conducted in the future.
Regarding the assessment of heart function and dimensions for multiple times, in our previous study we observed progressive ventricular dilatation over the time, but fractional shortening markedly dropped already on day 7 post-infarction with almost no further decline till day 28 [8]. This information is mentioned in the revised text.
- How do the authors control for the amount of injury sustained across the animals post MI surgery? The addition of a cohort of animals to assess the amount and variability of the initial tissue damage following the surgery is required.
To assess the amount and variability of the initial tissue damage following MI, additional groups being assessed immediately after ischemia/reperfusion insult are widely used. As indicted elsewhere, in this particular study we did not check for the extent of initial injury post-MI. However, we would like to mention at this point that coronary occlusion procedure has been routinely performed in our laboratory for years, repeatedly resulting in ischemia of around 40% of LV myocardium [2-4, 8, 9]. The occlusion lasting 60 min causes irreversible injury of the whole ischemic zone. To address this point, we modified the text accordingly. Please see Methods.
- The authors fail to account for the contribution of infiltrating cell types that occur post ischemic injury. Do the changes in the cell death markers observed correlate with changes in cell type content in the heart post injury?
This is an interesting point and the authors thank for such an important concept. Although it has not been our primary goal to focus and distinguish the either cell death mode in certain cardiac cells and non-cardiac cells (infiltrating immune cells, including neutrophils, monocytes/macrophages, dendritic cells and B and T lymphocytes) we have provided some statements to address this point in Discussion.
- Curtis, M. J.; Alexander, S.; Cirino, G.; Docherty, J. R.; George, C. H.; Giembycz, M. A.; Hoyer, D.; Insel, P. A.; Izzo, A. A.; Ji, Y.; MacEwan, D. J.; Sobey, C. G.; Stanford, S. C.; Teixeira, M. M.; Wonnacott, S.; Ahluwalia, A., Experimental design and analysis and their reporting II: updated and simplified guidance for authors and peer reviewers. Br J Pharmacol 2018, 175, (7), 987-993.
- Cervenka, L.; Huskova, Z.; Kopkan, L.; Kikerlova, S.; Sedlakova, L.; Vanourkova, Z.; Alanova, P.; Kolar, F.; Hammock, B. D.; Hwang, S. H.; Imig, J. D.; Falck, J. R.; Sadowski, J.; Kompanowska-Jezierska, E.; Neckar, J., Two pharmacological epoxyeicosatrienoic acid-enhancing therapies are effectively antihypertensive and reduce the severity of ischemic arrhythmias in rats with angiotensin II-dependent hypertension. J Hypertens 2018, 36, (6), 1326-1341.
- Neckar, J.; Hye Khan, M. A.; Gross, G. J.; Cyprova, M.; Hrdlicka, J.; Kvasilova, A.; Falck, J. R.; Campbell, W. B.; Sedlakova, L.; Skutova, S.; Olejnickova, V.; Gregorovicova, M.; Sedmera, D.; Kolar, F.; Imig, J. D., Epoxyeicosatrienoic acid analog EET-B attenuates post-myocardial infarction remodeling in spontaneously hypertensive rats. Clin Sci (Lond) 2019, 133, (8), 939-951.
- Lichy, M.; Szobi, A.; Hrdlicka, J.; Horvath, C.; Kormanova, V.; Rajtik, T.; Neckar, J.; Kolar, F.; Adameova, A., Different signalling in infarcted and non-infarcted areas of rat failing hearts: A role of necroptosis and inflammation. J Cell Mol Med 2019, 23, (9), 6429-6441.
- Muller-Edenborn, B.; Kania, G.; Osto, E.; Jakob, P.; Krasniqi, N.; Beck-Schimmer, B.; Blyszczuk, P.; Eriksson, U., Lidocaine Enhances Contractile Function of Ischemic Myocardial Regions in Mouse Model of Sustained Myocardial Ischemia. PLoS One 2016, 11, (5), e0154699.
- Barthel, H.; Ebel, D.; Mullenheim, J.; Obal, D.; Preckel, B.; Schlack, W., Effect of lidocaine on ischaemic preconditioning in isolated rat heart. Br J Anaesth 2004, 93, (5), 698-704.
- Kariya, N.; Cosson, C.; Mazoit, J. X., Comparative effect of lidocaine, bupivacaine and RAC 109 on myocardial conduction and contractility in the rabbit. Eur J Pharmacol 2012, 691, (1-3), 110-7.
- Hrdlicka, J.; Neckar, J.; Papousek, F.; Vasinova, J.; Alanova, P.; Kolar, F., Beneficial effect of continuous normobaric hypoxia on ventricular dilatation in rats with post-infarction heart failure. Physiol Res 2016, 65, (5), 867-870.
- Hrdlicka, J.; Neckar, J.; Papousek, F.; Huskova, Z.; Kikerlova, S.; Vanourkova, Z.; Vernerova, Z.; Akat, F.; Vasinova, J.; Hammock, B. D.; Hwang, S. H.; Imig, J. D.; Falck, J. R.; Cervenka, L.; Kolar, F., Epoxyeicosatrienoic Acid-Based Therapy Attenuates the Progression of Postischemic Heart Failure in Normotensive Sprague-Dawley but Not in Hypertensive Ren-2 Transgenic Rats. Front Pharmacol 2019, 10, 159.
Reviewer 2 Report
In this manuscript Lichy et al, observed markers of necroptosis, autophagy, and apoptosis in heart failure post ischemia reperfusion injury. This in vivo study utilized Wistar rats which were subjected to one hour ischemia followed by reperfusion and then cardiac function and markers of cell death were examined 6 weeks later. The authors report that all three cell death modalities have been implicated in the progression of heart failure and the purpose of their study was to expand the current knowledge about the relationship between necroptosis and autophagy and their specific roles in heart failure progression post myocardial infarction (MI) . The authors conclude that at 6 weeks post MI injury markers of necroptosis are increased, while markers of autophagy are not in the left ventricle. Thus, ruling out the possibility of autophagy being co-stimulated with necroptosis in dysfunctional cardiomyocytes. Additionally, hallmarks of apoptosis are significantly upregulated in the right ventricle but not the left at this time point. Suggesting that apoptosis may play a role in RV deterioration. Unfortunately, this manuscript is severely lacking the data needed to perform a conclusive analysis of the cell death that may occur in heart failure post MI. In its current form, the manuscript does not expand the current knowledge about the relationship between any cell death modalities taking place in heart failure.
Major Concerns
- The authors examine multiple cell death pathways by blotting for markers of cell death at one time point post cardiac injury. The authors fail to report histological analysis of the tissue examining the amount of cell death that is actually taking place at the time point selected.
- This study would have benefited from the addition of multiple time points to longitudinally assess markers of cell death that change during the progression of heart failure. In addition, cardiac function and dimensions should also be assessed multiple times to correlate with the changes in cell death markers.
- How do the authors control for the amount of injury sustained across the animals post MI surgery? The addition of a cohort of animals to assess the amount and variability of the initial tissue damage following the surgery is required.
- The authors fail to account for the contribution of infiltrating cell types that occur post ischemic injury. Do the changes in the cell death markers observed correlate with changes in cell type content in the heart post injury?
Round 2
Reviewer 2 Report
The authors failed to adequately address any of the reviewers concerns that required additional experimentation. In its updated form, the experimental design is still extremely lacking in this study. As is, no conclusions can be made about cell death during the progression of heart failure post MI.